# Peer J

# Integrative study of *Arabidopsis thaliana* metabolomic and transcriptomic data with the interactive MarVis-Graph software

Manuel Landesfeind[1], Alexander Kaever[1], Kirstin Feussner[2], Corinna Thurow[3], Christiane Gatz[3], Ivo Feussner[2] and Peter Meinicke[1]

[1] Department of Bioinformatics, Institute of Microbiology and Genetics, Georg-August-University, Göttingen, Germany
[2] Department for Plant Biochemistry, Albrecht-von-Haller-Institute for Plant Sciences, Georg-August-University, Göttingen, Germany
[3] Department for Plant Molecular Biology and Physiology, Schwann-Schleiden-Research-Center for Molecular Cell Biology, Georg-August-University, Göttingen, Germany

Corresponding author
Peter Meinicke, peter@gobics.de

## ABSTRACT

State of the art high-throughput technologies allow comprehensive experimental studies of organism metabolism and induce the need for a convenient presentation of large heterogeneous datasets. Especially, the combined analysis and visualization of data from different high-throughput technologies remains a key challenge in bioinformatics. We present here the MarVis-Graph software for integrative analysis of metabolic and transcriptomic data. All experimental data is investigated in terms of the full metabolic network obtained from a reference database. The reactions of the network are scored based on the associated data, and sub-networks, according to connected high-scoring reactions, are identified. Finally, MarVis-Graph scores the detected sub-networks, evaluates them by means of a random permutation test and presents them as a ranked list. Furthermore, MarVis-Graph features an interactive network visualization that provides researchers with a convenient view on the results. The key advantage of MarVis-Graph is the analysis of reactions detached from their pathways so that it is possible to identify new pathways or to connect known pathways by previously unrelated reactions. The MarVis-Graph software is freely available for academic use and can be downloaded at: http://marvis.gobics.de/marvis-graph.

## INTRODUCTION

High-throughput technologies notoriously generate large datasets often including data from different omics platforms. Each dataset contains data for several thousand experimental markers, e.g., mass-to-charge ratios in mass spectrometry or spots in DNA microarray analysis. An experimental marker is associated with an intensity profile which may include several measurements according to different experimental conditions (*Dettmer, Aronov & Hammock, 2007*). Usually, the intensity profiles are

normalized and tested for significant differences between the experimental conditions. In addition, the experimental markers can be analyzed by multivariate methods, such as cluster algorithms (*Golub et al., 1999*) or principal component analysis (*Alter, Brown & Botstein, 2000*). Significant differences or clusters may be explained by associated annotations, e.g., in terms of metabolic pathways or biological functions. During recent years, numerous specialized tools have been developed to aid biological researchers in automating all these steps (e.g., *Medina et al., 2010*; *Kaever et al., 2009*; *Wägele et al., 2012*). Comprehensive studies can be performed by combining technologies from different omics fields. The combination of transcriptomic and proteomic data sets revealed a strong correlation between both kinds of data (*Nie et al., 2007*) and supported the detection of complex interactions, e.g., in RNA silencing (*Haq et al., 2010*). Moreover, correlations were detected between RNA expression levels and metabolite abundances (*Gibon et al., 2006*). Therefore, tools that integrate, analyze and visualize experimental markers from different platforms are needed. To cope with the complexity of genome-wide studies, pathway models are utilized extensively as a simple abstraction of the underlying complex mechanisms. Set Enrichment Analysis (*Subramanian et al., 2005*) and Over-Representation Analysis (*Huang, Sherman & Lempicki, 2009*) have become state-of-the-art tools for analyzing large-scale datasets: both methods evaluate predefined sets of entities, e.g., the accumulation of differentially expressed genes in a pathway. Originally developed for genomic analyses (Gene Set Enrichment Analysis), Set Enrichment Analysis has also been applied as Metabolite Set Enrichment Analysis (*Xia & Wishart, 2010*) or Network-based Gene Set Enrichment Analysis (*Glaab et al., 2012*). Nevertheless, these approaches require a predefined grouping of the biological entities, e.g., into pathways, which always restricts the analysis to the known groups. While manually curated pathways are convenient and easy to interpret, experimental studies have shown that all metabolic and signaling pathways are heavily interconnected (*Kunkel & Brooks, 2002*; *Laule et al., 2003*). Data from biomolecular databases support these studies: the metabolic network of *Arabidopsis thaliana* in the KEGG database (*Kanehisa et al., 2012*; *Kanehisa & Goto, 2000*) contains 1606 reactions from which 1464 are connected in a single sub-network (>91%), i.e., they share a metabolite as product or substrate.[1] In the AraCyc 10.0 database (*Mueller, Zhang & Rhee, 2003*; *Rhee et al., 2006*), more than 89% of the reactions are counted in a single sub-network. In both databases, most other reactions are completely disconnected. Additionally, Set Enrichment Analyses can not identify links between the predefined sets easily. This becomes even more important when analyzing smaller pathways as provided by the MetaCyc (*Caspi et al., 2008*; *Caspi et al., 2012*) database. Moreover, methods that utilize pathways as predefined sets ignore reactions and related biomolecular entities (e.g., metabolites, genes) which are not associated with a single pathway. For example, this affects 4000 reactions in MetaCyc and 2500 in KEGG, respectively (*Altman et al., 2013*). Therefore, it is desirable to develop additional methods that do not require predefined sets but may detect enriched sub-networks in the full metabolic network.

Currently, to our knowledge, there is no tool available to incorporate experimental markers from different high-throughput experiments and to relate them in the context

[1] KEGG release 64.0, downloaded on October 25th 2012 via the KEGG API.

of metabolic reaction-chains. While several tools support the statistical analysis of experimental markers from one or more omics technologies and then utilize variants of Set Enrichment Analysis (*Xia et al., 2012*; *Chen et al., 2013*; *Howe et al., 2011*), no tool is able to explicitly search for connected reactions that include most of the metabolites, genes, and enyzmes with experimental evidence. However, the automatic identification of sub-networks has been proven useful in other contexts, e.g., in the analysis of protein–protein-interaction networks (*Alcaraz et al., 2012*; *Baumbach et al., 2012*; *Maeyer et al., 2013*).

The presented MarVis-Graph software aims to close this gap. MarVis-Graph imports experimental markers from different high-throughput experiments and analyses them in the context of reaction-chains in full metabolic networks. Then, MarVis-Graph scores the reactions in the metabolic network according to the number of associated experimental markers and identifies sub-networks consisting of subsequent, high-scoring reactions. The resulting sub-networks are ranked according to a scoring method and visualized interactively. Hereby, sub-networks consisting of reactions from different pathways may be identified to be important whereas the single pathways may not be found to be significantly enriched. MarVis-Graph may also connect reactions without an assigned pathway to reactions within a particular pathway. The MarVis-Graph tool was applied in a case-study investigating the wound response in *Arabidopsis thaliana* to analyze combined metabolomic and transcriptomic high-throughput data.

## MATERIALS AND METHODS

MarVis-Graph analyses experimental markers from one or more high-throughput experiments and different omics platforms in the context of a full metabolic network. Metabolic networks can be created from the KEGG database (*Kanehisa et al., 2012*; *Kanehisa & Goto, 2000*) and the BioCyc collection (*Caspi et al., 2012*; *Caspi et al., 2008*). Different datasets are integrated into a metabolic network, reactions are scored using the associated experimental data and connected high-scoring reactions (sub-networks) are then identified. The sub-networks are ranked according to a scoring method, evaluated utilizing a random permutation test, and visualized within an interactive graphical user interface.

### Representing metabolic networks

To represent full metabolic networks, especially in combination with experimental data, a flexible and expandable data structure is required. MarVis-Graph models the metabolic network as an undirected graph in which each entity (a molecule, a reaction, an enzyme, etc.) is represented by a single vertex. Mathematically, the graph $G = (V, L)$ consists of vertices $V$ and edges $L = V \times V$. The vertices represent the following groups of biochemical entities:

- $M$: experimental markers for metabolites (metabolite markers),
- $C$: metabolites,
- $R$: reactions,
- $E$: enzymes,

- $H$: genes,
- $T$: experimental markers for transcripts (transcript markers),
- $P$: pathways.

A minimal example of a graph containing only one vertex of each subset is shown in Fig. 1.

An edge between two vertices $v_1, v_2$ is added to the graph if the vertices are related within the metabolic network, e.g., a metabolite is a substrate or product of a reaction. Edges are represented as tuple $l = (v_1, v_2)$ and the set of edges is restricted to

$$L = \left\{ l : l = (v_i, v_j) \in (M \times C) \cup (C \times R) \cup (R \times E) \cup (E \times H) \cup (H \times T) \cup (R \times P) \right\}.$$

For example, metabolite markers are only connected to metabolites while in return metabolites can only be connected to metabolite markers and reactions (see Fig. 1).

### *Generation of metabolic networks*

MarVis-Graph has a built-in support for the creation of metabolic networks from the KEGG database and the BioCyc collection (source databases). The networks can either consist of all reactions from a database (reference network) or only reaction from a specific organism. Data from the KEGG database is downloaded directly via the KEGG API (*KEGG, 2013*). Databases from the BioCyc collection have to be downloaded as zipped TAR archives (*BioCyc, 2013*) which are then imported into MarVis-Graph. MarVis-Graph saves and loads generated metabolic networks utilizing a simple XML format.

### Experimental markers

Experimental markers can be imported from tabular data files, e.g., comma-separated-values (.csv) or Microsoft® Excel® spreadsheet format (.xls, .xlsx) files (for an example see Data S2). In general, an experimental marker in MarVis-Graph consists of a unique identifier (ID) and a weight that represents its quality or importance (see below). Additionally, an experimental marker can contain an intensity or abundance profile where each intensity is labeled with a condition-name. MarVis-Graph does not use these abundances in the analysis but merely visualizes them in the context of the identified sub-networks. The mapping of the experimental markers to corresponding biological entities in the metabolic network (*marker annotation*) may also be provided.

A critical step is the calculation of weights for the experimental markers because of their influence to the extraction of sub-networks (see "Identification of sub-networks"). Usually, statistical analysis tools calculate a $p$-Value for each experimental marker based on its abundances in the different conditions of the experiment. For simplicity, weights for MarVis-Graph can be calculated based on these $p$-Values with

$$weight_m = 1 - p\text{-}Value_m.$$

Using external tools beforehand, MarVis-Graph can utilize the existing expertise in a particular omics field where the experimental markers originate from. If no weight is given, MarVis-Graph will assume a default of 1.

**Figure 1 Schema of the metabolic network representation in MarVis-Graph.** Metabolite markers are shown in gray, metabolites in red, reactions in blue, enzymes in green, genes in yellow, transcript markers in pink, and pathways in turquoise color. The edges are shown in black with labels that comply with the biological meaning. The orange arrows depict the flow of score for the initial scoring (described in section "Initial Scoring").

### Metabolite markers

In MarVis-Graph, metabolite markers obtained from mass-spectrometry experiments additionally contain the experimental mass. The experimental mass has to be calculated based on the mass-to-charge ratio ($m/z$-value) and specific isotope- or adduct-corrections (*Draper et al., 2009*) by means of specialized tools, e.g., MarVis-Filter (*Kaever et al., 2012*).

If no marker annotation is given for the imported metabolite markers, MarVis-Graph utilizes the masses to map the metabolite markers to metabolites based on a simple mass-comparison. To cope with measurement errors, the tolerance of the mapping has to be specified (default $0.005u$). Note that matching metabolite markers to metabolites by mass-comparison is error-prone and may result in many false matches. Therefore, it is preferable to calculate the exact compound using sophisticated technologies (*Dunn et al., 2013*) and provide this information during the import.

### Transcript markers

For each transcript marker the corresponding annotation has to be given. In DNA microarray experiments, each spot (transcript marker) is specific for a gene and can therefore be used for annotation. For other technologies an annotation has to be provided by external tools.

## Identification of sub-networks

In MarVis-Graph, each reaction is scored initially based on the associated experimental data (see "Initial scoring"). This initial scoring is refined (see "Refining the scoring") and afterwards reactions with a score below a user-defined threshold are removed. The network is decomposed into subsequent high-scoring reactions that constitute the sub-networks.

### Initial scoring

The weight of each experimental marker (see "Experimental markers") is equally distributed over all metabolites and genes associated with the metabolite marker or transcript marker, respectively. For all vertices, this is repeated as illustrated in Fig. 1 until the weights are accumulated by the reactions.

### Refining the scoring

Datasets from high-throughput technologies might not cover all metabolites or transcripts, e.g., if

- they are not measurable: specific metabolites may not be detected by mass spectrometry analysis. Furthermore, several metabolites exist only for a short period of time within a protein complex that catalyze more than one reaction step,
- they are filtered out by statistical analysis: transcripts may be equally expressed throughout all experimental conditions if the corresponding products are required all the time. This is especially true for enzymes that are not rate-limiting. The amount of a metabolite might not change across the different conditions when it is metabolized immediately.

If a reaction has associated metabolites or enzymes for which no or only a few experimental markers have been detected, the reaction will receive a very low initial reaction score. But, reactions with a low score that connect reactions with high scores should be considered in the sub-network extraction. To cope with these gaps, the *random walk with restart* (RWR) algorithm (*Yin et al., 2010*) is applied to distribute parts of the reaction score to the neighboring reactions. For an efficient calculation, a graph consisting only of reactions (*reaction graph*) is constructed from the metabolic network. Two reactions in this graph are connected if they share a metabolite as substrate or product. Often, reactions are connected solely by "hub metabolites", e.g., *NADP* and *ATP*, that take part in a high number of reactions (*Faust & van Helden, 2012*). However, these connections via hub metabolites are not informative and should be ignored. A general definition of a hub metabolite is not possible because it highly depends on the source database. For a user-defined parameter $c$, a metabolite is considered a hub metabolite if it contributes in $c$ or more reactions as a substrate or product. Edges in the reaction graph that were added only because of hub metabolites are removed.

### Calculating sub-networks

The initial reaction scores are used as input scoring for the random walk algorithm. The algorithm is performed as described by *Glaab et al. (2012)* with a user-defined restart-probability $r$ (default value 0.8). After convergence of the algorithm, reactions with a score lower than the user-defined threshold $t$ (default value $t = 1 - r$) are removed from the reaction network. During the removal process, the network is decomposed into pairwise disconnected sub-networks containing only high-scoring reactions.

In the following, a resulting sub-network is denoted by a prime: $G' = (V', L')$ with $V' = M' \cup C' \cup R' \cup E' \cup G' \cup T' \cup P'$.

## Ranking of sub-networks

The identified sub-networks are ranked with one of the following scoring methods and presented in a sorted list.

### Graph size

The size of a sub-network is the total number of contained reactions:

$$s_\text{s}(G') = |R'|$$

### Graph diameter

The graph diameter is an intuitive description of the dimension of a network. For a given sub-network $G'$, the diameter is the maximum distance between all pairs of reactions $r_i, r_j \in R'$ whereas the distance $d(r_i, r_j)$ is the length of the shortest path:

$$s_\text{d}(G') = \max_{r_i, r_j \in R'} \{d(r_i, r_j)\}.$$

### Sum of weights

The weights of the experimental markers are distributed to the reactions as described in the previous section. The sub-network score is the sum of the reaction scores in the sub-network:

$$s_\text{sow}(G') = \sum_{r \in R'} score(r).$$

## Evaluation of the ranking

The scores of the identified sub-networks can be assessed using a random permutation test, evaluating the marker annotations under the null hypothesis of being connected randomly. Here, the assignments from metabolite markers to metabolites and from transcript markers to genes are randomized. For each association between a metabolite marker and a metabolite, this connection is replaced by a connection between a randomly chosen metabolite marker and a randomly chosen metabolite. The random metabolite marker is chosen from the pool of formerly connected metabolite markers. Each connected transcript marker is associated with a randomly chosen gene. Choosing from the list of already connected experimental markers ensures that the sum of weights from the original and the permuted network are equal. This method differs from the commonly utilized *XSwap* permutation (*Hanhijärvi, Garriga & Puolamäki, 2009*) that is based on swapping endpoints of two random edges. The main difference of our permutation method is that it results in a network with different topological structure, i.e., different degree of the metabolite and gene nodes. However, when all experimental markers have equal weight (see "Experimental markers") the XSwap method would result in exactly the same network and therefore is not applicable here.

Finally, the sub-networks are detected and scored with the same parameters applied for the original network. Based on the scores of the networks identified in the random permutations, the family-wise-error-rate (FWER) and false-discovery-rate (FDR) are calculated for each originally identified sub-network.

## RESULTS AND DISCUSSION

MarVis-Graph was applied in a case study investigating the *A. thaliana* wound response. Data from a metabolite fingerprinting (*Meinicke et al., 2008*) and a DNA microarray experiment (*Yan et al., 2007*) were imported into a metabolic network specific for *A. thaliana* created from the AraCyc 10.0 database (*Lamesch et al., 2011*). The metabolome and transcriptome have been measured before wounding as control and at specific time points after wounding in wild-type and in the *allene oxide synthase* (*AOS*) knock-out mutant *dde-2-2* (*Park et al., 2002*) of *A. thaliana Columbia* (see Table 1). The AOS mutant was chosen, because AOS catalyzes the first specific step in the biosynthesis of the hormone jasmonic acid, which is the key regulator in wound response of plants (*Wasternack & Hause, 2013*).

### Preprocessing of the datasets

Both datasets have been preprocessed with the MarVis-Filter tool (*Kaever et al., 2012*) utilizing the Kruskal–Wallis *p*-value calculation on the intensity profiles. Based on the ranking of ascending *p*-values, the first 25% of the metabolite markers and 10% of the transcript markers have been selected for further investigation (Data S2). The filtered metabolite and transcript markers were imported into the metabolic network. For metabolite markers, metabolites were associated if the metabolite marker's detected mass differs from the metabolites monoisotopic mass by a maximum of $0.005u$. Transcript markers were linked to the genes whose ID equaled the ID given in the CATMA database (*Sclep et al., 2007*) for that transcript marker. Table 2 lists the numbers of reactions, metabolites, enzymes, and genes as well as metabolite and transcript markers in the final metabolic network. For this evaluation, all experimental markers were imported into MarVis-Graph with a default weight of 1. Because of the former filtering of the experimental markers, all were assumed to be of equal interest for the analysis.

### Resulting sub-networks

The MarVis-Graph algorithm involves several parameters (see "Identification of sub-networks") that cannot directly be estimated. In this case-study, the parameters for MarVis-Graph were set to

- restart-probability $r = 0.8$.

  The restart-probability of the RWR algorithm controls the amount of score that is distributed equally to the neighboring reactions, i.e.,

$$(1 - r) \times score.$$

  The higher the restart-probability the more the algorithm emphasizes near neighbors in the network. With a low restart-probability, the score is distributed widely over the network and may connect usually disconnected sub-networks.

- score-threshold $t = 0.2$.

  The score-threshold determines the reactions that are considered for sub-network construction after performing the RWR algorithm and directly depends on the weights

**Table 1 Samples in the experimental datasets.** Number of DNA microarray and metabolic mass spectrometry samples (biological and technical replicates) at different time points (hpw: hours past wounding, M: metabolic data samples, T: transcriptomics data samples).

| Time point | 0 hpw | | 0.5 hpw | | 1 hpw | | 2 hpw | | 5 hpw | |
|---|---|---|---|---|---|---|---|---|---|---|
| | M | T | M | T | M | T | M | T | M | T |
| wild type | 9 | 7 | 9 | - | 9 | 3 | 9 | - | 9 | - |
| dde-2-2 | 9 | 7 | 9 | - | 9 | 3 | 9 | - | 9 | |

**Table 2 Vertices in the *A. thaliana* specific metabolic network after import of experimental markers.** Number of objects in the metabolic network in absolute counts and relative abundances. For experimental markers, the with annotation column gives the number of metabolite markers and transcript markers that were annotated with a metabolite or gene, respectively. The direct evidence column contains the number of metabolites and genes, that are associated with a metabolite marker or transcript marker. For enzymes, this is the number of enzymes encoded by a gene with direct evidence. The number of vertices with an association to a reaction is given in the with reaction column. In the last column, this is given for associations to metabolic pathways.

| | Overall | With annotation | | Direct evidence | | With reaction | | With pathway | | |
|---|---|---|---|---|---|---|---|---|---|---|
| | | Count | Percent | Count | Percent | Count | Percent | Count | Percent (overall) | Percent (with reaction) |
| Metabolite markers | 12030 | 697 | 5.79% | | | 532 | 4.42% | 524 | 4.36% | 98.50% |
| Transcript markers | 2538 | 825 | 32.51% | | | 710 | 27.97% | 376 | 14.81% | 52.96% |
| Metabolites | 3310 | | | 564 | 17.04% | 2383 | 71.99% | 1914 | 57.82% | 80.32% |
| Genes | 6895 | | | 803 | 11.65% | 5811 | 84.28% | 2610 | 37.85% | 44.91% |
| Enzymes | 7130 | | | 802 | 11.25% | 6017 | 84.39% | 2806 | 39.35% | 46.63% |
| Reactions | 3542 | | | | | | | 2056 | 58.05% | |

of the experimental markers given on import. When using a restart-probability of 0.8, a score-threshold of 0.2 keeps only nodes that are high-scoring by themselves, have a very high-scoring neighbor, or are enclosed by several high-scoring neighbors (see Fig. F5).

- hub metabolite-threshold $c = 10$.

 The hub metabolite-threshold was chosen based on expert knowledge: a threshold of 10 was just low enough to eliminate known hub metabolites, e.g., ATP, in the AraCyc database.

Based on these settings, MarVis-Graph detected a total of 133 sub-networks. The sub-networks were ranked according to size $S_s$, diameter $S_d$, and sum-of-weights $S_{sow}$ scores (Table S4). Interestingly, the different rankings show a high correlation with all pairwise correlations higher than 0.75 (Pearson correlation coefficient) and 0.6 (Spearman rank correlation).

### Allene-oxide cyclase sub-network

In all rankings, the sub-network *allene-oxide cyclase* (named after the reaction with the highest score in this sub-network) appeared as top candidate. Therefore, it was investigated

further and discussed in detail in this study. This sub-network is constituted of reactions from different pathways related to fatty acids. Figure 2 shows a visualization of the sub-network.

*Jasmonic acid biosynthesis.* The main part of the sub-network is formed by reactions from the "jasmonic acid biosynthesis" (*Plant Metabolic Network, 2013*) resulting in *jasmonic acid* (*jasmonate*). The presence of this pathway is very well established because of its central role in mediating the plants wound response (*Reymond & Farmer, 1998*; *Creelman, Tierney & Mullet, 1992*). Additionally, metabolites and transcripts from this pathway were expected to show prominent expression profiles because *AOS*, a key enzyme in this pathway, is knocked-out in the mutant plant.

*Jasmonic acid derivatives and hormones.* *Jasmonate* is a precursor for a broad variety of plant hormones (*Wasternack & Hause, 2013*), e.g., the derivative *(-)-jasmonic acid methyl ester* (also *Methyl Jasmonic Acid*; MeJA) is a volatile, airborne signal mediating wound response between plants (*Farmer & Ryan, 1990*).

Reactions from the *jasmonoyl-amino acid conjugates biosynthesis I* (*PMN, 2013a*) pathway connect *jasmonate* to different amino acids, including *L-valine*, *L-leucine*, and *L-isoleucine*. Via these amino acids, this sub-network is connected to the *indole-3-acetyl-amino acid biosynthesis* (*PMN, 2013b*) (IAA biosynthesis). Again, this pathway produces a well known plant hormone: Auxine (*Woodward & Bartel, 2005*). Even though, *jasmonate* and *auxin* are both plant hormones, their connection in this subnetwork is of minor relevance because amino acid conjugates are often utilized as active or storage forms of signaling molecules. While jasmonoyl-amino acid conjugates represent the active signaling form of jasmonates, IAA amino acid conjugates are the storage form of this hormone (*Staswick et al., 2005*).

*Poly-hydroxy fatty acids biosynthesis.* Besides being the precursor for *jasmonate*, $\alpha$-*linolenate* may be metabolized to fatty acid derivatives containing an epoxide group. Up to now, the function of the poly–hydroxy derivatives of $\alpha$-*linolenate* is not known. However, the epoxide containing derivative of *linoleate* is known to have a function in plant defense (*Hou & Forman III, 2000*).

*Traumatin biosynthesis.* The first reaction of the *jasmonic acid biosynthesis*, transforming $\alpha$-*linolenate* to *13(S)-hydroperoxy-9(Z),11(E),15(Z)-octadecatrienoate*, is shared with the *traumatin biosynthesis*. In the *traumatin biosynthesis*, *linoleate* (see paragraph "Poly-hydroxy fatty acids biosynthesis") is degraded too. Both, *linoleate* and $\alpha$-linolenate, are metabolized to *traumatin* and the reactions are likely to occur in parallel because *13S-lipoxygenase* enzymes catalyze both reactions.

$\Delta^{12}$-*fatty acid dehydrogenase.* The enzyme $\Delta^{12}$-*fatty acid dehydrogenase*[2] does not exists in *A. thaliana*, but the $\omega$-*3-fatty acid desaturase* (*PMN, 2013c*) is annotated with the same enzymatic classification number 1.14.99.33 (*ExPASy, 2013*) as it catalyzes the same reaction on other reactants. The former name is the accepted name for the reaction by the enzyme

______________________________

[2] *Available at http://pmn.plantcyc.org/ ARA/NEW-IMAGE?&object=1.14.99. 33-RXN (accessed 14 May 2013).*

**Figure 2 Schema of the allene-oxide cyclase sub-network.** Metabolites are show in red, reactions in blue, and enzymes in green color. Metabolites and reactions without direct experimental evidence are marked by a dashed outline and a brighter color while enzymes without experimental evidence are hidden. The metabolic pathways described in section "Resulting sub-networks" are highlighted with different colors. The orange and green parts indicate the reaction chains required to build *jasmonate* and its amino acid conjugates. The coloring of pathways was done manually after export from MarVis-Graph.

3 *Available at http://pmn.plantcyc.org/ ARA/NEW-IMAGE?&object=1.14.99. 33-RXN* (accessed 14 May 2013).

commission and therefore used by the AraCyc database[3] from which this metabolic network was built. Furthermore, the reaction's product, *crepenynic acid*, does not exists. The *ω-3-fatty acid desaturase* should catalyze a reaction from *linoleate* to *α-linolenate*. Metabolite markers that match the mass of *crepenynic acid* do also match *α-linolenate* because both molecules have the same sum-formula and monoisotopic mass.

As mentioned above, MarVis-Graph compiled the metabolic network for this study from the AraCyc database version 10.0. On June 4th, a curator changed the database to remove the $\Delta^{12}$-*fatty acid dehydrogenase* prior to the release of AraCyc version 11.0.

## CONCLUSION

The presented new software tool MarVis-Graph supports the investigation and visualization of omics data from different fields of study. The introduced algorithm for identification of sub-networks is able to identify reaction-chains across different pathways and includes reactions that are not associated with a single pathway. The application of MarVis-Graph in the case study on *A. thaliana* wound response resulted in a convenient graphical representation of high-throughput data which allows the analysis of the complex dynamics in a metabolic network.

## AVAILABILITY

The MarVis-Graph tool is implemented in Java 7, released under the terms of the GPL v3.0 (*Free Software Foundation, Inc., 2007*) and can be used free of charge in academic research. Although MarVis-Graph can be well integrated into the work-flow of the MarVis-Suite it is only available as a stand-alone tool. MarVis-Graph can be obtained from the supplemental material (Data O1) or by download from: http://marvis.gobics.de/ marvis-graph.

## ACKNOWLEDGEMENTS

We like to thank Kathrin P. Aßhauer for fruitful discussions.

### Funding

This work has partially been funded by the German Federal Ministry of Education and Research (BMBF 0315595A) and the German Research Council (DFG). Alexander Kaever and Manuel Landesfeind were supported by the Biomolecules program of the Göttingen Graduate School for Neurosciences, Biophysics, and Molecular Biosciences (GGNB). The funders had no role in study design, data collection and analysis, decision to publish, or preparation of the manuscript.

### Grant Disclosures

The following grant information was disclosed by the authors:
German Federal Ministry of Education and Research: 0315595A.
German Research Council.

## Competing Interests

Ivo Feussner is an Academic Editor for PeerJ. We declare no further competing interests.

## Author Contributions

- Manuel Landesfeind conceived and designed the experiments, performed the experiments, analyzed the data, wrote the paper.
- Alexander Kaever conceived and designed the experiments, performed the experiments, contributed reagents/materials/analysis tools, wrote the paper.
- Kirstin Feussner and Ivo Feussner conceived and designed the experiments, performed the experiments, analyzed the data, contributed reagents/materials/analysis tools, wrote the paper.
- Corinna Thurow and Christiane Gatz analyzed the data, contributed reagents/materials/analysis tools.
- Peter Meinicke conceived and designed the experiments, performed the experiments, wrote the paper.

## Supplemental Information

Supplemental information for this article can be found online at http://dx.doi.org/10.7717/peerj.239.

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
