# Peer review of "Integrative study of Arabidopsis thaliana metabolomic and transcriptomic data with the interactive MarVis-Graph software"

_PeerJ, doi:10.7717/peerj.239_

## Round 0.1 · original submission · Minor Revisions

· Academic Editor

Minor Revisions

The manuscript appears well written and introduces new methods of sorting and visualizing datasets to assess metabolic pathway features. This approach helps to resolve relationships in pathways that may exist apart from the static formats often seen in other pathway representations. Both reviewers brought up significant points which may improve the manuscript; please review their suggestions and try to rebuild and integrate them into a revised version as best as possible. Please make sure the 'SEA' term is used properly. Since the use of mass-spectroscopy data in tandem with transcriptome data appears to be a popular comparison addressing the reviewer's comments is this area would be useful. This may help ease some of the readers ability to address such matters when adopting this approach.

With numerous technologies developing over the years to interpret biological systems in different ways, the utility of this approach can help establish a formal approach toward identifying alternate mechanisms. It appears from the figures provided that use of the software works in building significant pathways, but from the website presentation only partial MarVis-Suite components were seen; is the Marvis-Graph portion going to be integrated with this, or exist as a separate entity which works with MarVis-Suite software. Thank you for submitting this manuscript and I expect it to be well received. Congratulations on your efforts. Apologies for the delays caused by some setbacks beyond my control.

Additional suggested edits:

Example of annotation:
LINE/PAGE NO.: / PREVIOUS FORM / SUGGESTED FORM / [ADDITIONAL NOTES]

Abstr.: / hight-hroughput / high-throughput / []
Abstr.: / studies of an organisms metabolism / studies of organism metabolism / []
Abstr.: / omics / “omics” / [please refine use of this term. 'omics may also work.]
Abstr.: / Bioinformatics / bioinformatics / [does not need to be capitalized.]
Abstr.: / reactions are identified / reactions which are identified / []
Pg 3. : / usefull in other context. / useful in other contexts. / []
Pg 3. : / ?Maeyer, Renkens, / (Maeyer, Renkens, / [parenthesis needed here.]
Pg 4. : / reactions are scored based / reactions are scored and based / []
Pg 4. : / (sub-networks) are identified / (sub-networks) are then identified / []
Pg 5. : / (.csv), Microsoft . . . / (.csv), or other MS Excel spreadsheet formats (.xls, .xlsx) . . . / [simplify trademark and versions usage here.]
Pg 6. : / accumulated by the reactions / accumulated by the parsed reactions / []
Pg 6. : / or more reactions as substrate / or more reactions as a substrate / []
Pg 8. : / AOS knock-out / allene oxide synthase (AOS) knock-out / [define AOS.]
Pg 9. : /// [when mentioning allene-oxide cyclase you may want to also indicate how AOS is related.]
Pg 9. : / AOS / AOS / [can leave as is if previous edit made to define AOS.]

·

Basic reporting

The manuscript follows the required standard of PeerJ.

Experimental design

The authors used data acquired from experiments already published, therefore the experimental design is well chose to test the applicability and demonstrate the power of the described software.

Validity of the findings

The data are already published, so they are robust and therefore a good case to be used to test the new software. The networks established by the software and described in this manuscript are sufficiently described.

Additional comments

This manuscript addresses an important gap and issue currently faced in the 'omics sciences and that is to have a computational platform to integrate data sets derived from different 'omics experiments. Although there is some software available to map changes determined from 'omics experiments, this software described in the manuscript allows to determine network structures based on biochemical knowledge (publically available metabolic pathways) and overlay those with actual measured data.
I have only one comment, and this is regarding matching accurate mass data directly to metabolites. There are very many metabolites which have exactly the same mass (isobaric). How does the proposed software deal with most likely miss identifications? The matching of metabolite markers (M) with metabolites (C) does not consider retention time or mass fragmentation data to have a greater confidence that the matching is right and the right C will be chosen for the M measured. Metabolite identifications from LC-MS metabolite profiling analysis using accurate mass as the only feature for identification is almost impossible and error rate is massive. Therefore building metabolite networks based on potentially inaccurate or error-prone ID can lead to wrong interpretations and conclusions. If the authors could include paragraphs in the manuscript how this issue can be dealt with better would definitely enhance the manuscript.

Reviewer 2 ·

Basic reporting

I noticed a few minor issues in the discussion of previous methods in the Introduction: In the field of pathway enrichment analysis the abbreviation SEA usually stands for "Singular enrichment analysis" not "Set Enrichment Analysis" (see distinction between SEA, GSEA and MEA in Huang et al., Bioinformatics enrichment tools: paths toward the comprehensive functional analysis of large gene lists, Nucl. Acids Res., 2009). MSEA and NSEA do not belong to the cut-off based SEA approaches, because they involve cut-off independent algorithms (NSEA belongs to the group of network-based / modular enrichment analysis methods, while MSEA provides access to multiple algorithms falling into different categories). I would recommend that the authors use the existing and commonly used category definitions (see Huang et al. paper mentioned above) rather than using new definitions for already existing abbreviations, which may confuse some of the readers.
When criticizing that previous approaches "require a predefined grouping of biological entities, e.g. into pathways" and proposing a network-based approach, the authors should also take into consideration that manually curated pathway definitions are typically easier to interpret than uncurated genome-scale networks. Indeed, for their own example analysis presented in the Results section the authors also interpret their top-candidate sub-network by decomposing it into different pathways (see page 9, "This sub-network is constituted of reactions from different pathways related to fatty acids..."). Thus, there is a trade-off between the advantages of genome-scale networks in terms of comprehensiveness of information and the benefits of pathways in terms of interpretability, and the approach proposed by the authors rather belongs to the methods making use of both sources of information for the final interpretation than being completely pathway-independent.

Experimental design

Overall, the experimental design is clearly described and logical, only a few aspects of the methodology were not clear to me based on the text.

Section 2.2 says that the score for a marker represents its quality or importance (by default 1), but how is the quality or importance for a measured metabolite or transcript quantified specifically as a deviation from the default score 1? The same section mentions that "abundances are not used in the analysis but may be visualized", however, since DNA microarray measurements and mass spectrometry measurements ultimately reflect (relative) transcript/metabolite abundances, which other information is used for defining a marker score if not abundance-based information? Fold changes and significance-of-differential expression scores for array data also depend on the comparison of relative mRNA abundances between sample groups, hence, it should be clarified how a meaningful transcript score can be defined without taking the indirectly measured mRNA abundances into account.

The authors also mention that the proposed approach depends on three parameters, which have been set to certain default values for the experiments described in the Results section (r = 0.8, t = 0.2, c = 10). However, it is not clear how these parameters were chosen and how strongly the results depend on these parameters. For the default parameters, the authors obtain 133 sub-networks scored as significant, but would this number change significantly when modifying the parameters and how should the reader choose the parameter values when applying the approach on a new dataset?

Validity of the findings

The methodology proposed by the authors, using biological network information to integrate different omics data sources and a graph-based statistical analysis to score the enrichment of sub-networks in important transcriptomics/metabolomics markers, contains new ideas and should in principle provide useful biological results. However, I see a potential issue with regard to the permutation-based evaluation of sub-networks scores. The authors randomly re-assign edges between metabolite markers and metabolites, and transcript markers and genes, respectively, which does not necessarily preserve the topology of the original graph. Identified sub-networks in the original graph may therefore appear more significant when compared to the permutation-based sub-networks, because in this background model typical topological properties of biological networks may not be retained (the degree distribution in biological graphs tends to differ from the distribution observed in random networks, see Jeong et al., The large-scale organization of metabolic networks, Nature, 2000). However, it should be possible to resolve this potential issue by using random edge-swap perturbations (i.e. selecting pairs of edges randomly and swapping their endpoints, see S. Hanhijärvi et al., Randomization techniques for graphs, Proceedings of the Ninth SIAM International Conference on Data Mining, 2009; or alternatively, Li and Kurata, Visualizing Global Properties of Large Complex Networks, PLoS ONE, 2008).

---

## Round 0.2 · accepted · Accept

· Academic Editor

Accept

Thank you for taking the time to address suggested revisions. The revised version reads very smoothly and concerns from the initial review are adequately addressed. The manuscript was in now in good form for publication. Though you have a statement relating to this tool as being a first to relate different high-throughput experiments for metabolic reaction chains, I am reminded of a pathway tool (Letovsky) incorporated into the ACEDB software (Durbin and Thierry-Mieg, 1994) that may have had the capability to address some of these issues since entries are treated in an object-oriented manner. Nevertheless, this is a modern day application to address current data needs and flexible data sources; I applaud your efforts and anticipate the use of these methods to be applied to new and complex metabolomic investigations. Congratulations.

I would consider this version acceptable for publication; however, please address these last minor modifications:

Example of annotation:
LINE/PAGE NO.: / PREVIOUS FORM / SUGGESTED FORM / [ADDITIONAL NOTES]

Section 1: /(?)/(nn)/ [please include a citation for this reference for cluster algorithms.]
Section 1: /e.g. (... / (... / [not sure you really need to subset these citations as examples.]
Section 1: /A. thaliana/Arabidopsis thaliana/ [first usage in manuscript should include full genus-species representation; last usage in the section may be abbreviated.]
Section 3.2 /// [Adding parameter notes was a nice addition.]